# Scene Graph Generation with Role-Playing Large Language Models

**Guikun Chen**[1*], **Jin Li**[3*], **Wenguan Wang**[1,2†]

[1]Zhejiang University

[2]National Key Laboratory of Human-Machine Hybrid Augmented Intelligence, Xi'an Jiaotong University

[3]Changsha University of Science & Technology

https://github.com/guikunchen/SDSGG

## Abstract

Current approaches for open-vocabulary scene graph generation (OVSGG) use vision-language models such as CLIP and follow a standard zero-shot pipeline – computing similarity between the query image and the text embeddings for each category (*i.e.*, text classifiers). In this work, we argue that the text classifiers adopted by existing OVSGG methods, *i.e.*, category-/part-level prompts, are *scene-agnostic* as they remain unchanged across contexts. Using such fixed text classifiers not only struggles to model visual relations with high variance, but also falls short in adapting to distinct contexts. To plug these intrinsic shortcomings, we devise SDSGG, a *scene-specific* description based OVSGG framework where the weights of text classifiers are adaptively adjusted according to the visual content. In particular, to generate comprehensive and diverse descriptions oriented to the scene, an LLM is asked to play different roles (*e.g.*, biologist and engineer) to analyze and discuss the descriptive features of a given scene from different views. Unlike previous efforts simply treating the generated descriptions as *mutually equivalent* text classifiers, SDSGG is equipped with an advanced *renormalization* mechanism to adjust the influence of each text classifier based on its relevance to the presented scene (this is what the term "*specific*" means). Furthermore, to capture the complicated interplay between subjects and objects, we propose a new lightweight module called mutual visual adapter. It refines CLIP's ability to recognize relations by learning an interaction-aware semantic space. Extensive experiments on prevalent benchmarks show that SDSGG outperforms top-leading methods by a clear margin.

## 1  Introduction

SGG [1] aims to create a structured representation of an image by identifying objects as nodes and their relations as edges within a graph. The emerging field of OVSGG [2, 3], which broadens the scope of SGG to identify and associate objects beyond a predefined set of categories, has become a research hotspot for its prospective to amplify the practicality of SGG in diverse real-world applications.

OVSGG has achieved notable progress due to the success of vision-language models (VLMs) [4, 5] and prompt learning [6, 7]. Existing OVSGG methods adopt a standard zero-shot pipeline [4], which computes similarity between the visual embedding from query image and the text embedding from pre-defined text classifiers (*c.f*. Fig. 1a). One straightforward direction for OVSGG is to use only the category name (*e.g.*, "riding") [2, 8, 9, 3, 10] as the text classifier and perform vision-language

---

* The first two authors contribute equally to this work.

† Corresponding Author: Wenguan Wang.

38th Conference on Neural Information Processing Systems (NeurIPS 2024).

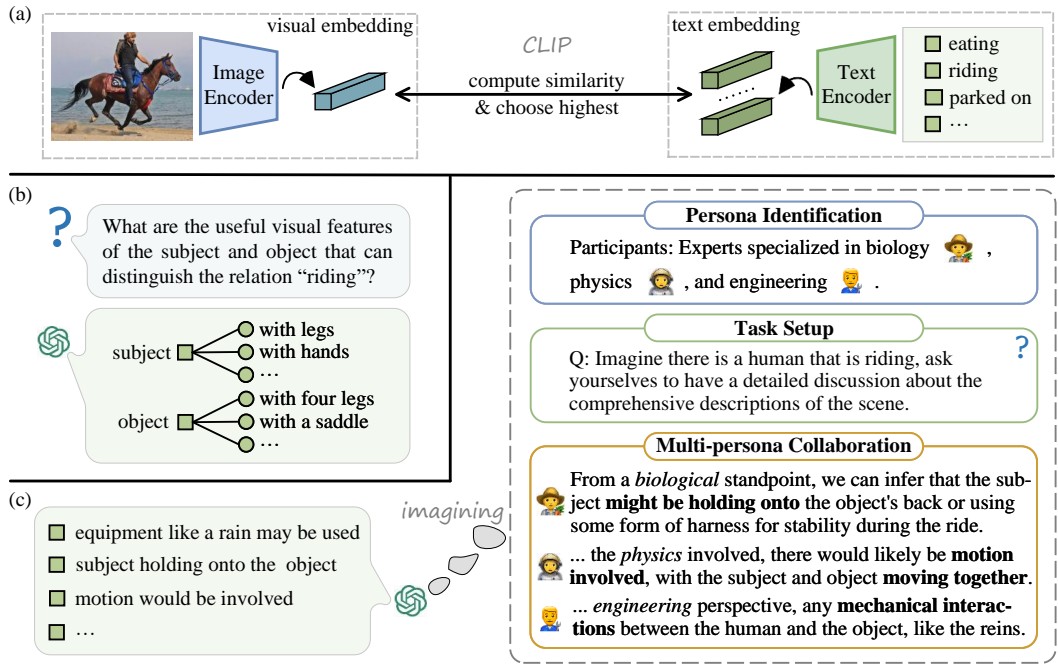

Figure 1: Illustration of the used text classifiers in OVSGG. (a) CLIP performs zero-shot classification by computing similarity between the query image and the text embeddings for each category, then choosing the highest. (b) To further utilize the learned semantic space of CLIP, one can compute similarities of multiple part-level prompts (*e.g.*, the object of ⟨man, riding, horse⟩ may be described with "with four legs" and "with a saddle"). (c) Instead of using these *scene-agnostic* text classifiers, SDSGG adopts comprehensive, *scene-specific* descriptions generated by LLMs, which can adapt to specific contexts by using the proposed renormalization.

alignment as in prompt learning to learn the underlying patterns. On the other hand, [11, 12] argue that such methods fail to utilize the rich context of additional information that language can provide. To address this, [12] decomposes relation detection into several *separate* components, computing similarities by checking how well the visual features of the object match the part-level descriptions[1]. As shown in Fig. 1b, the object of relation "riding" should have four legs, a saddle, *etc*.

Despite these technological advances, we still observe current OVSGG systems lack in-depth inspection of the expressive range of the used text classifiers, which puts a performance cap on them. Concretely, OVSGG models that rely on *scene[2]-agnostic* text classifiers have the following flaws. **First**, methods based on category names [2, 3] struggle to model the large variance in visual relations. Using only category names as classifiers [4] does hold water when applied to object recognition [13]. For instance, CLIP shares common visual features across diverse image-text pairs of tigers which encompasses a variety of tiger appearances and corresponding descriptions. Nonetheless, the scenario becomes far more complex when it comes to relation detection. The visual features that define the relation "on" can vary dramatically across scenes, *e.g.*, "dog on bed" *vs*. "people on road". RECODE [12] proposes to decompose relation detection into recognizing part-level descriptions for both subject and object, hence partially easing the aforementioned difficulty. Yet, it computes similarities for the subject and object *separately* and does not model the *interplay* between subjects and objects. **Second**, part-level prompt based methods uniformly process all descriptions as affirmative classifiers [11, 12], overlooking the possibility that some text classifiers might be contrary to specific contexts. When querying LLMs for distinctive visual features of subjects and objects to distinguish the predicate "riding", with the subject as "human" and the object as "horse", LLMs provide part-level descriptions of the expected appearance of both entities. All generated descriptions are treated equally as definitive text classifiers. However, these descriptions could potentially be misleading, as LLMs produce them without considering the specific context, even resulting in some descriptions that are wholly irrelevant to the presented image. For example, LLMs typically associate the predicate "riding" with the animal

---

[1]The terms "description" and "prompt" are used interchangeably, all denoting text classifiers.

[2]A scene typically involves objects and their interplay and relationships.

"with four legs". Nonetheless, such associations are indeed irrelevant, as an animal's legs are not always visible in the presented image, or the animal may have only two legs.

Filling the gaps identified above calls for a fundamental paradigm shift: moving from the fixed, *scene-agnostic* (*i.e.*, category-/part-level) text classifiers towards flexible, *scene-specific* ones. In light of this, we develop SDSGG, a scene-specific description based OVSGG framework that utilizes text classifiers generated from LLMs, complemented by a renormalization technique, to understand scenes from different perspectives. **For the textual part**, given a scene with specified content, an LLM is assigned distinct roles, akin to experts specializing in biology, physics, and engineering, to analyze descriptive scene features comprehensively (*c.f.* Fig. 1c). Such a multi-persona scheme is designed to improve the diversity of the generated scene descriptions as LLMs tend to generate repetitive content. LLM can be queried multiple times to obtain a large number of scene descriptions. Moreover, since not all descriptions are relevant to the presented image (*e.g.*, some parts of the object may not appear), SDSGG is equipped with an advanced mechanism that renormalizes each scene description via opposite descriptions corresponding to the original descriptions. This involves evaluating two vision-language similarities: one for the original scene description and the other for its opposite. The *difference* between the two similarities is viewed as the *self-normalized* similarity of the scene description, allowing for flexible control over its influence. For instance, an irrelevant description would yield a *self-normalized* similarity close to zero, as the two similarity scores of it and its opposite would be very close. By doing so, the generated scene-level descriptions become flexible, *scene-specific* descriptions (SSDs). **For the visual part**, we propose a new adapter for relation detection, called mutual visual adapter, which consists of several lightweight learnable modules. The proposed adapter projects CLIP's semantic embeddings into another interaction-aware space, modeling the complicated interplay between the subject and object through cross-attention.

With the proposed adaptive SSDs, our SDSGG is capable of: **i**) adapting to the given context via evaluating the *self-normalized* similarity of each SSD; **ii**) alleviating the overfitting problem in OVSGG models [2, 3] that use only one classifier; and **iii**) naturally generalizing to novel relations by associating them with SSDs. We validate SDSGG on two widely-used benchmarks, *i.e.* Visual Genome (VG) [14] and GQA [15]. Experimental results show that SDSGG outperforms existing OVSGG methods [3] by a large margin. The strong generalization and promising performance of SDSGG evidence the great potential of *scene-specific* description based relation detection.

## 2   Related Work

**Scene Graph Generation.** Since [1] introduces iterative message passing for SGG, research studies in structured visual scene understanding have witnessed phenomenal growth. Tremendous progress has been achieved and can be categorized into: **i**) Two-stage SGG [16, 17, 18, 19, 20], which first detects all objects in the images and then recognizes the pairwise relations between them; **ii**) Debiased SGG [21, 22, 23, 24, 25, 26, 27, 28], which focuses on the problem of long-tailed predicate distribution in the current dataset; **iii**) Weakly-supervised SGG [29, 30, 31, 32, 33], which investigates how to generate scene graph with only image-level supervision; **iv**) One-stage SGG [34, 35, 36, 37, 38, 39], which implements SGG within an end-to-end framework (also exemplified in other relation detection tasks [40, 41, 42]), discarding several hand-crafted procedures; **v**) Open-vocabulary SGG, which learns to recognize unseen categories during training by using category-level [2, 8, 9, 3, 10] or part-level prompts [12]. **vi**) Few-show SGG, which learns to recognize relations given a few examples [43].

Existing OVSGG frameworks adopt a standard open-vocabulary learning paradigm, *i.e.*, perform vision-language alignment in the pre-trained or random initialized semantic space with supervision of only the category names. One except [12] reformulates OVSGG from recognizing category-level prompts into recognizing part-level prompts, by decomposing SGG into several separate components and computing their similarities independently, in a training-free manner. SDSGG represents the best of both worlds. On the one hand, we point out the drawbacks of the commonly used *scene-agnostic* text classifiers and introduce *scene-specific* alternates to understand scenes from different perspectives. On the other hand, SDSGG incorporates a learnable mutual visual adapter to capture the underlying patterns in the dataset and proposes to renormalize text classifiers for adapting to specific contexts.

**Open-vocabulary Learning.** Most deep neural networks operate on the close-set assumption, which can only identify pre-defined categories that are present in the training set. Early zero-shot learning approaches [44, 45, 46] adopt word embedding projection to constitute the classifiers for

unseen class classification. With the rapid progress of vision language pre-training [4, 5, 47], open vocabulary learning [48] has been proposed and demonstrates impressive capabilities by, for example, distilling knowledge from VLMs [49, 50, 51, 52, 53, 54], exploiting caption data [55, 56], generating pseudo labels [57, 58, 59, 60, 61], training without dense labels [62, 63, 64], joint learning of several tasks [65, 66, 67], and training with more balanced data [68, 69].

While sharing a very high-level idea of vision-language alignment in open-vocabulary methods, our SDSGG **i**) explicitly models the context-dependent scenarios and introduces *scene-specific* text classifiers as the flexible learning targets, and **ii**) incoreporates a new mechanism for computing *self-normalized* similarities, thereby renormalizing text classifiers according to the presented image.

**VLMs Meet LLMs [70].** The big win for VLMs has been all about getting the model to match up pictures and their descriptions closely while keeping the mismatched ones apart [4, 5, 47]. This trick, inspired by contrastive learning from self-supervised learning [71, 72, 73, 74, 75], helps VLMs get really good at figuring out what text goes with what image. Moreover, prompt learning acts as a flexible way to communicate with VLMs, giving them a nudge or context to apply their knowledge of images and text in specific ways [6, 76, 77, 7, 78]. In addition to hand-crafted or learnable prompts, [11] offers a fresh perspective, *i.e.*, using LLMs to generate detailed, comprehensive prompts as the inputs of VLMs' text encoder. Many follow-up works [79, 80, 81, 82, 83, 84, 85] across various domains and tasks demonstrate the effectiveness of integrating VLMs and LLMs.

Category-/part-level prompts are *scene-agnostic* and cannot adapt to specific contexts. To this end, SDSGG adopts *scene-specific* descriptions, generated by LLMs in a multi-persona collaboration fashion, as the inputs of CLIP's text encoder. Different from part-level prompt based approaches [11, 12] which processes all part-level prompts as affirmative classifiers, SDSGG provides a flexible alternative via the association between classifiers (*i.e.*, SSDs) and categories, and the renormalizing strategy *w.r.t.* each SSD. Since the learned semantic space of VLMs may not be sensitive to relations [12], we design a lightweight mutual visual adapter to project them into interaction-aware space for capturing the complicated interplay between the subject and object.

# 3   Methodology

**Task Setup and Notations.** Given an image $I$, SGG transforms it into a structured representation, *i.e.*, a directed graph $\mathcal{G} = \{\mathcal{O}, \mathcal{R}\}$, where $\mathcal{O}$ represents localized (*i.e.*, bounding box) objects with object category information and $\mathcal{R}$ represents pairwise relations between objects. For a fair comparison, this work focuses on predicting $\mathcal{R}$ given $\mathcal{O}$, *i.e.*, the predicate classification task which avoids the noise from object detection, as suggested by [1, 3, 12]. Our study delves into the intricacies of transitioning SGG from a traditional closed-set setting to an open vocabulary paradigm. This transition enables the system to recognize previously unseen predicate categories (*i.e.*, `novel` split) by learning from observed predicate categories (*i.e.*, `base` split) during training.

SDSGG follows the standard zero-shot pipeline with VLMs [4], which computes similarity between the visual embedding $v$ and the text embedding $t$ for each category, and the category with highest similarity is viewed as the final classification result (*c.f*. Fig. 2a). For each subject-object pair, $v$ can be derived by feeding cropped patches from the input image $I$ into the visual encoder. The text embedding $t$ used in existing OVSGG frameworks falls into two main settings: **i**) Each category consists of only one text classifier, *i.e.*, the category name itself. **ii**) Each category consists of multiple text classifiers w.r.t. subject and object, *i.e.*, part-level descriptions. SDSGG reformulates the text classifiers as *scene-specific* descriptions which will be detailed in §3.1.

**Algorithmic Overview.** SDSGG is a SSD based framework for OVSGG, supported by the cooperation of VLMs and LLMs. For the textual part (*c.f*. Fig. 2b), SDSSG enjoys the expressive range of the comprehensive SSDs generated by LLMs' multi-persona collaboration. This is complemented by a renormalizing mechanism to adjust the influence of each text classifier. For the visual part (*c.f*. Fig. 2c), SDSSG is equipped with a mutual visual adapter to aggregate visual features $v$ from $I$ for a given subject-object pair. After introducing how we generate and use SSDs for the text part (§3.1) and the mutual visual adapter for interplay modeling of the subject and object (§3.2), we will elaborate on SDSGG's training objective (§3.3).

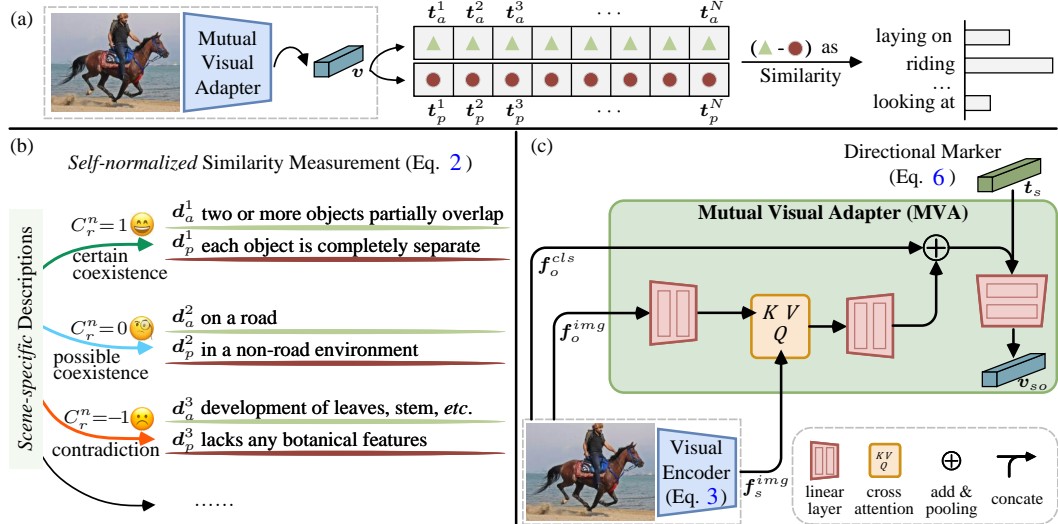

Figure 2: (a) Overview of SDSGG. (b) Each text classifier of SDSGG contains a raw description $\boldsymbol{d}_a^n$ and an opposite description $\boldsymbol{d}_p^n$. As such, the *self-normalized* similarities can be computed with the association ($C_r^n$) between predicate categories and SSDs. (c) Given the visual features (*i.e.*, $\boldsymbol{f}_s^{img}$, $\boldsymbol{f}_o^{cls}$, and $\boldsymbol{f}_o^{img}$) of both the subject and object extracted from CLIP's visual encoder, our mutual visual adapter (MVA) projects them into interaction-aware space and models their complicated interplay with cross-attention.

## 3.1 Scene-specific Text Classifiers

**Scene-level Description Generation via Multi-persona Collaboration.** Using standard prompts to query LLMs is a direct way to generate scene descriptions. For instance, one could straightforwardly prompt LLM with a question like "Imagine there is an animal that is eating, what should the scene look like?". LLM's response would typically sketch out an envisioned scene based on its statistical training on large corpus. However, these responses may not fully capture the scene's complexity, often overlooking aspects such as the spatial arrangement of elements and the background environment.

To alleviate this, we draw inspiration from recent advances in LLMs' multi-persona capabilities [86, 87, 88, 89]. Specifically, LLM adopts three distinct roles, mirroring the expertise found in experts specializing in biology, physics, and engineering. This approach allows for a comprehensive discussion of what a given scene entails. Because each query to LLM usually only yields 3-5 sentences of description, we query LLM several times, each time giving LLM a different scene content to be discussed, thus obtaining a large number of scene descriptions. Since these initial descriptions may suffer from noise and semantic overlap, we ask LLM to streamline and combine these descriptions, ensuring more cohesive and distinct scene-level descriptions $\mathcal{D}_l = \{d^1, d^2, \cdots, d^N\}$ and corresponding text embeddings $\mathcal{T} = \{\boldsymbol{t}^1, \boldsymbol{t}^2, \cdots, \boldsymbol{t}^N\}$, where $N$ denotes the number of SSDs and text embeddings $\mathcal{T}$ are extracted by the text encoder of CLIP. Due to the limited space, we provide more details and prompts for generating scene descriptions in the appendix (§D).

**Association between Scene-level Descriptions and Relation Categories.** So far, we have obtained various scene-level descriptions that have the ability to represent diverse scenes. A critical inquiry arises regarding their utility for relation detection, given their lack of explicit association with specific relation categories. To address this, we delineate three distinct scenarios characterizing the interplay between relation categories and scene descriptions: **i**) certain coexistence ($C_r^n = 1$), where a direct correlation exists; **ii**) possible coexistence ($C_r^n = 0$), indicating a potential but not guaranteed association; and **iii**) contradiction ($C_r^n = -1$), denoting an incompatibility between the scene description and relation category. Here $C_r^n$ denotes the correlation between relation $r \in \mathcal{R}$ and $n_{th}$ scene description, and is generated by LLMs (prompts are shown in §D). Such a categorization enables us to calculate the similarity for each relation category:

$$sim(\boldsymbol{v}, r) = \sum\nolimits_{n=1}^{N} C_r^n * \langle \boldsymbol{v}, \boldsymbol{t}^n \rangle, \qquad (1)$$

where $\langle \cdot, \cdot \rangle$ denotes the cosine similarity with temperature [4].

**Scene-specific Descriptions = Scene-level Descriptions + Reweighing.** Upon examining text classifiers in depth, we noticed that certain classifiers are contextually bound, *e.g.*, "two or more objects partially overlap each other" may not exist in all scenes. This observation underscores the necessity for a mechanism to evaluate the significance of each text classifier, rather than applying a uniform weight across the board. This is exactly what the term "*specific*" means. Recall that the similarity measurement between an image and the text "a photo of a cat" alone yields limited insight. However, when juxtaposed with multiple texts, such as "a photo of a cat/dog/tiger", the comparison of similarity scores across these categories reveals which category (cat, dog, or tiger) the image most closely resembles. Inspired by this, we propose the incorporation of an opposite description $d_p^n$ (§D) for each raw scene description $d_a^n$ as a reference point (*e.g.*, "two or more objects partially overlap each other" *vs.* "each object is completely separate with clear space between them"), resulting in SSDs $\mathcal{D}_s = \{(d_a^1, d_p^1), (d_a^2, d_p^2), \cdots, (d_a^N, d_p^N)\}$ and updated text embeddings $\mathcal{T} = \{(\boldsymbol{t}_a^1, \boldsymbol{t}_p^1), (\boldsymbol{t}_a^2, \boldsymbol{t}_p^2), \cdots, (\boldsymbol{t}_a^N, \boldsymbol{t}_p^N)\}$. Subsequently, the *self-normalized* similarity is defined as:

$$sim(\boldsymbol{v}, r) = \sum_{n=1}^{N} C_r^n * (\langle \boldsymbol{v}, \boldsymbol{t}_a^n \rangle - \langle \boldsymbol{v}, \boldsymbol{t}_p^n \rangle). \tag{2}$$

The difference in similarity scores, *i.e.*, $\langle \boldsymbol{v}, \boldsymbol{t}_a^n \rangle - \langle \boldsymbol{v}, \boldsymbol{t}_p^n \rangle$, quantifies the relative contribution of that SSD. By such means, a SSD irrelevant to the presented context will have a minimal effect, as the similarity scores of it ($\langle \boldsymbol{v}, \boldsymbol{t}_a^n \rangle$) and its opposite ($\langle \boldsymbol{v}, \boldsymbol{t}_p^n \rangle$) would be nearly identical.

## 3.2 Mutual Visual Adapter

After introducing how to obtain the text embeddings and how to compute vision-language similarity, one question remains at this point: how to obtain visual embeddings? When given a subject-object pair with bounding boxes from $\mathcal{O}$, there exist various strategies for aggregating visual features for the subject and object within $\boldsymbol{I}$. For example, traditional closed-set SGG frameworks [16, 17] employ RoI pooling to extract visual features for specified bounding boxes, subsequently fusing these features for further classification. In contrast, the more recent OVSGG framework [12] uses the visual encoder of CLIP to extract visual embeddings of both subject and object. Then, it processes two visual embeddings *independently* through part-level descriptions. Such an independent approach, however, overlooks the informative interplay between the subject and object.

To address this oversight and capture the complicated interactions between subject and object, we introduce a new component: the mutual visual adapter (MVA). MVA is composed of several lightweight, learnable modules designed to fine-tune CLIP's visual encoder specifically for pairwise relation detection. This approach aims to enhance the model's ability to recognize the nuanced interactions that define relationships between subject and object in an image.

**Regional Encoder.** Given an image $\boldsymbol{I}$ and a subject-object pair with bounding boxes ($b_s$ and $b_o$) from $\mathcal{O}$, the initial visual embeddings can be obtained from the visual encoder of CLIP:

$$\boldsymbol{f}_{s/o} = [\boldsymbol{f}_{s/o}^{cls} | \boldsymbol{f}_{s/o}^{img}] = [\boldsymbol{f}_{s/o}^{cls} | \boldsymbol{f}_{s/o}^1, \boldsymbol{f}_{s/o}^2, \cdots, \boldsymbol{f}_{s/o}^M] = \texttt{Encoder}_v\big(\texttt{Crop}(\boldsymbol{I}, b_{s/o})\big), \tag{3}$$

where $M$ denotes the number of patches, $\texttt{Encoder}_v$ is CLIP's visual encoder that is kept frozen during training, and $\texttt{Crop}$ represents image cropping.

**Visual Aggregator.** Next, MVA is adopted to aggregate $\boldsymbol{f}_s$ and $\boldsymbol{f}_o$ by cross-attention and two lightweight projection modules. Let the subject part be the query, and the object part be the key and value. The patch embeddings of object $\boldsymbol{f}_o^{img}$ are first projected into low-dimensional, semantic space:

$$\boldsymbol{l}_o^{img} = \texttt{Linear}_{down}(\boldsymbol{f}_o^{img}), \tag{4}$$

where $\texttt{Linear}$ denotes a standard fully connected layer. Afterwards, cross-attention is adopted to capture the complicated interplay between subject and object, resulting in an aggregated visual embedding for the given subject-object pair:

$$\boldsymbol{v}_{so} = \texttt{Linear}_{up}\big(\texttt{AvgPool}(\texttt{LN}(\boldsymbol{f}_o^{cls} + \texttt{CrossAttn}(\boldsymbol{f}_s^{img}, \boldsymbol{l}_o^{img})))\big), \tag{5}$$

where $\texttt{AvgPool}$ is the average pooling. $\texttt{LN}$ is the standard layer normarlization. $\texttt{CrossAttn}(\boldsymbol{Q}, \boldsymbol{KV})$ denotes the standard cross-attention operation. $\boldsymbol{v}_{os}$ can be computed in a similar way by exchanging the query and key of cross-attention. Combining them together leads to the final visual embedding $\boldsymbol{v} = (\boldsymbol{v}_{so} + \boldsymbol{v}_{os})/2$ for final similarity measurement. As such, MVA captures the interplay of subject and object in the projected, interaction-aware space.

**Directional Marker.** One may notice that the structure of MVA is *symmetric* and has no information about which input branch is the subject/object. This has a relatively small effect on semantic relations, but a significant effect on geometric relations. For instance, after exchanging the location of the subject and object (image flipping), the relation "eating" remains unchanged, while the relation "on the left" would become "on the right". Here we simply incorporate two text embeddings ($t_s$ and $t_o$) of "a photo of subject/object" into MVA and thus update the visual embedding as:

$$v_{so} = \texttt{Linear}\big(\texttt{Concate}(v_{so}, t_s)\big), \tag{6}$$

where $\texttt{Concate}$ denotes concatenation. $v_{os}$ and $v$ can be updated accordingly. Further exploration of directional marker, *e.g.*, incorporating more complex feature fusion modules, is left for future work.

### 3.3 Training Objective

A typical training objective in open-vocabulary learning aims to bring representations of positive pairs closer and push negative pairs apart in the embedding space. In SDSGG, the term "positive/negative pairs" is not defined at the category level *but at the description level*, requiring losses tailored for different relation-description association types. Given a labeled relation triplet, one simplest contrastive loss can be defined as:

$$\mathcal{L} = \frac{1}{|\mathcal{T}|} \sum_{t^n \in \mathcal{T}} \big( \underbrace{\langle v, t_a^n \rangle - \langle v, t_p^n \rangle}_{similarity} - \underbrace{\alpha * C_r^n}_{target} \big)^2, \tag{7}$$

where $\alpha$ is a scaling factor. However, as for scene descriptions marked by possible coexistence (*i.e.*, $C_r^n = 0$), there is no direct target that can be used for training. Inspired by the identical mapping in residual learning [90], we make the prediction results of MVA close to those of CLIP. As such, MVA can learn the implicit knowledge embedded in CLIP's semantic space. In addition, this regularization term prevents MVA from overfitting to relations in the $\texttt{base}$ split, which is a common problem in open-vocabulary learning. Hence, the loss is further reformulated as:

$$\mathcal{L} = \frac{1}{|\mathcal{T}|} \sum_{t^n \in \mathcal{T}} \big( \underbrace{\langle v, t_a^n \rangle - \langle v, t_p^n \rangle}_{similarity} - \underbrace{\alpha * C_r^n}_{target} - \underbrace{(\beta * sim_{CLIP}(I, rel) - \lambda)}_{margin} \big)^2, \tag{8}$$

where $\beta$ is another scaling factor, $sim_{CLIP}(I, rel)$ denotes the vision-language similarity derived from the original CLIP, and $\lambda$ is a constant scalar and is empirically set to 3e-2.

## 4 Experiment

### 4.1 Experimental Setup

**Dataset.** We evaluate our method on GQA [15] and VG [14] following [3, 12].

**Split.** Following previous work [3], VG is divided into two splits: $\texttt{base}$ and $\texttt{novel}$ split. The $\texttt{base}$ split comprises 70% of the relation categories for training, while the $\texttt{novel}$ split contains the remaining 30% categories invisible during training. For a more comprehensive comparison, we also conduct testing on the $\texttt{semantic}$ set, encompassing 24 predicate categories [16, 12] with richer semantics. $\texttt{base}$ and $\texttt{novel}$ split of GQA [15] are obtained in a similar manner (§B).

**Evaluation Metrics.** We report Recall@K (R@K) and Recall@K(mR@K) following [3, 22].

**Base Models and Competitors.** As for the $\texttt{base}$ and $\texttt{novel}$ split, we compare SDSGG with two baselines: **i**) CLS [4], which uses only the category-level prompts to compute the similarity between the image and text; and **ii**) Epic [3], a latest OVSGG method, which introduces an entangled cross-modal prompt approach and learns the cross-modal embeddings using contrastive learning. In terms of the $\texttt{semantic}$ split, we compare our SDSGG with three baselines: **i**) CLS [4], which uses only the category-level prompts; **ii**) CLSDE [12], which uses prompts of relation class description; and **iii**) RECODE [12], which uses visual cues of several separate components. Since [10] has neither released the detailed split nor the code, it is not included in the comparisons for fairness.

**Implementation Details.** Due to limited space, implementation details are left in the appendix (§B).

### 4.2 Quantitative Comparison Result

We conduct quantitative experiments on VG [14] and GQA [15]. To ensure the performance gain is reliable, each experiment is repeated three times. The average and standard deviation are reported.

Table 1: Quantitative results (§4.2) on VG [14] `base` and `novel`.

| Method | Split | R@20↑ | R@50↑ | R@100↑ | mR@20↑ | mR@50↑ | mR@100↑ |
|---|---|---|---|---|---|---|---|
| CLS[ICML21][4] | | 2.1 | 3.2 | 3.9 | 7.0 | 9.0 | 10.9 |
| Epic[ICCV23][3] | `base` | - | 22.6 | 27.2 | - | - | - |
| **Ours** | | $18.7_{\pm 0.69}$ | $26.5_{\pm 0.92}$ | $31.6_{\pm 1.00}$ | $9.2_{\pm 0.14}$ | $12.4_{\pm 0.12}$ | $14.8_{\pm 0.10}$ |
| CLS[ICML21][4] | | 13.2 | 18.1 | 22.2 | 11.5 | 17.9 | 23.8 |
| Epic[ICCV23][3] | `novel` | - | 7.4 | 9.7 | - | - | - |
| **Ours** | | $18.4_{\pm 0.53}$ | $25.4_{\pm 0.48}$ | $29.6_{\pm 0.42}$ | $17.1_{\pm 0.42}$ | $25.2_{\pm 0.95}$ | $31.2_{\pm 1.09}$ |

Table 2: Quantitative results (§4.2) on VG [14] `semantic`.

| Method | R@20↑ | R@50↑ | R@100↑ | mR@20↑ | mR@50↑ | mR@100↑ |
|---|---|---|---|---|---|---|
| CLS[ICML21][4] | 7.2 | 10.9 | 13.2 | 9.4 | 14.0 | 17.6 |
| CLSDE[NeurIPS23][12] | 7.0 | 10.6 | 12.9 | 8.5 | 13.6 | 16.9 |
| RECODE†[NeurIPS23][12] | 7.3 | 11.2 | 15.4 | 8.2 | 13.5 | 18.3 |
| RECODE[NeurIPS23][12] | 9.7 | 14.9 | 19.3 | 10.2 | 16.4 | 22.7 |
| RECODE⋆[NeurIPS23][12] | 10.6 | 18.3 | 25.0 | 10.7 | 18.7 | 27.8 |
| **Ours** | $21.5_{\pm 0.47}$ | $29.3_{\pm 0.53}$ | $34.9_{\pm 0.66}$ | $16.8_{\pm 0.08}$ | $22.7_{\pm 0.41}$ | $28.4_{\pm 0.67}$ |

**VG [14]** `base` **and** `novel`. Table 1 illustrates our compelling results over the latest OVSGG model, Epic [3]. Since Epic did not release the full code, we report values under the same setting as in the original paper. CLS [4] leverages the original CLIP's checkpoint (*w/o* fine-tuning) and only the category-level prompts. The performance of CLS on the `novel` split is better than that on the `base` split because **i**) the `base` split contains more relation candidates to be classified (35 *vs*. 15); and **ii**) the `base` split contains head, uninformative relations (*e.g.*, "on" and "has") which are hard to be distinguished by CLIP [12]. By perform contrastive learning on the `base` split to learn the entangled cross-modal prompt, Epic [3] achieves significant performance gains on the `base` split (*i.e.*, 22.6% *vs*. 3.2% R@50 and 27.2% *vs*. 3.9% R@100). However, Epic demonstrates worse performance on the `novel` split (*i.e.*, 7.4% *vs*. 18.1% R@50 and 9.7% *vs*. 22.2% R@100), which indicates its overfitting on the training data. By incorporating SSDs, SDSGG outperforms Epic by a large margin on both the `base` split and `novel` split. For instance, SDSGG exceeds Epic by **3.9%**/**4.4%** R@50/100 on the `base` split and **18.0%**/**19.9%** R@50/100 on the `novel` split, respectively. The significant performance improvements on the `novel` set demonstrate the strong zero-shot capability of our approach. In addition, our margins over the CLS are **16.6%**∼**27.7%** R@K and **2.2%**∼**3.9%** mR@K on the `base` split, and **5.2%**∼**7.3%** R@K and **5.6%**∼**7.4%** mR@K on the `novel` split, respectively.

**VG [14]** `semantic`. In Table 2, we present the numerical results of SDSGG against the latest OVSGG work [12] on the `semantic` split. By leveraging part-level prompts, RECODE demonstrates superior performance compared to CLS [4] and CLSDE [12]. The introduction of filtering strategies, *i.e.*, RECODE⋆, shows substantial improvements. As seen, SDSGG surpasses all counterparts with remarkable gains on all metrics. In particular, SDSGG exceeds RECODE⋆ by **10.9%**/**11.0%**/**9.9%** on R@20/50/100, and **6.1%**/**4.0%**/**0.6%** on mR@20/50/100. Notably, SDSGG achieves impressive performance without relying on additional augmentation or data [10]. Since an increase on mR@K implies the average of the improvements for all categories, yields on mR@K may be relatively lower than that on R@K. Without bells and whistles, SDSGG establishes a new *state-of-the-art*.

**GQA [15]** `base` **and** `novel`. Since the codes of Epic [3] and RECODE [12] are insufficient to support replication, we only compare SDSGG with CLS on GQA [15]. As shown in Table 3, SDSGG outperforms CLS by a large margin across all splits and metrics. Since CLS is not trained on the `base` split, its performance on R@K is rel-

Table 3: Quantitative results on GQA [14] `base` & `novel`.

| Method | Split | R@20/50/100 | mR@20/50/100 |
|---|---|---|---|
| CLS [4] | | 4.2 / 6.4 / 7.9 | 8.9 / 13.2 / 15.3 |
| **Ours** | `base` | **33.4 / 43.9 / 49.5** | **15.6 / 21.0 / 24.5** |
| | | $\pm 1.10 / \pm 1.21 / \pm 1.20$ | $\pm 0.47 / \pm 0.38 / \pm 0.79$ |
| CLS [4] | | 21.3 / 28.3 / 32.1 | 16.6 / 27.0 / 29.4 |
| **Ours** | `novel` | **27.2 / 37.4 / 42.9** | **23.8 / 32.8 / 37.3** |
| | | $\pm 0.35 / \pm 0.58 / \pm 0.33$ | $\pm 0.21 / \pm 1.35 / \pm 1.33$ |

atively low due to the massive annotations of geometric relations that are not semantically rich [12].

Taking together, our extensive results provide solid evidence that SDSGG successfully unlocks the power of LLMs in OVSGG, and SSD is a promising alternative to category-/part-level prompts.

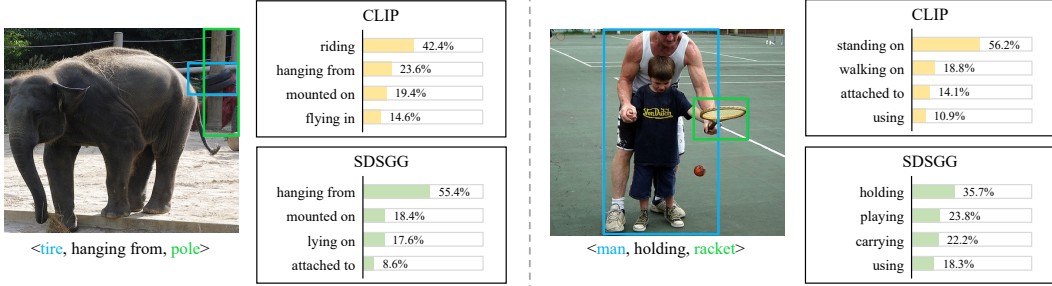

Figure 3: Visual results (§4.3) on VG [14].

## 4.3 Qualitative Comparison Result

Fig. 3 visualizes qualitative comparisons of SDSGG against CLIP [4] on VG [14]. As seen, with the proposed SSDs, SDSGG can generate higher-quality relation predictions even in challenging scenarios. We respectfully refer the reviewer to the appendix (§E) for more qualitative comparisons.

## 4.4 Diagnostic Experiment

For thorough evaluation, we conduct a series of ablative studies on VG [14].

**Textual Part.** We first study the effectiveness of our multi-persona collaboration (MPC) for *scene-specific* description generation (§3.1) in Table 4. Here we use the standard prompts to query LLMs for generating descriptions. As seen, without MPC, the performance drops drastically,

Table 4: Ablation studies (§4.4) on MPC.

| Method | Split | R@20/50/100 | mR@20/50/100 |
|---|---|---|---|
| **Ours** | base | **18.7 / 26.6 / 31.6** | **9.1 / 12.3 / 14.7** |
| *w/o* MPC | | 6.0 / 9.3 / 11.8 | 2.5 / 5.1 / 7.6 |
| **Ours** | novel | **18.9 / 25.8 / 30.0** | **16.6 / 25.2 / 31.5** |
| *w/o* MPC | | 5.2 / 8.7 / 11.8 | 4.9 / 11.0 / 16.2 |

*e.g.*, 11.8%/11.8% *vs.* 31.6%/30.0% R@100 on the `base`/`novel` split, respectively. This indicates the importance of the text classifiers used as they have impact on both training and testing.

While the effectiveness of MPC has been validated, one may wonder: **i**) Why use these three roles? **ii**) How to ensure the completeness and quality of generated classifiers? We want to highlight that: **i**) Different roles are used to increase the variety of descriptions. There is no word on exactly which roles should be used. **ii**) These open problems are beyond the scope of this work. We leave them for future work. **iii**) This work makes the first attempt to enhance classifier generation for OVSGG via MPC. **iv**) The experimental results suggest that the current generated SSDs are *good enough* for commonly used relation categories. To provide more empirical results, we investigate the impact of the proposed *self-normalized* similarities and the number of used SSDs in the appendix (§A).

**Visual Part.** Then, we examine the impact of mutual visual adapter (MVA, §3.2) and directional marker (DM, §3.2) in Table 5. The 1st row denotes a strong baseline, *i.e.*, a multi-layer perceptron with comparable parameters to aggregate the visual features. Upon projecting the visual features into interaction-aware space and applying cross-

Table 5: Ablation studies (§4.4) on the visual part.

| MVA | DM | Split | R@20/50/100 | mR@20/50/100 |
|---|---|---|---|---|
| | | base | 13.1 / 18.7 / 22.4 | 8.6 / 11.6 / 13.8 |
| ✓ | | | 17.0 / 23.8 / 28.2 | **9.2** / 12.3 / 14.6 |
| ✓ | ✓ | | **18.7 / 26.6 / 31.6** | 9.1 / **12.3 / 14.7** |
| | | novel | 17.4 / 24.7 / 28.9 | 17.2 / **26.5** / 30.9 |
| ✓ | | | 18.6 / 25.1 / 28.7 | **17.4** / 25.0 / 31.0 |
| ✓ | ✓ | | **18.9 / 25.8 / 30.0** | 16.6 / 25.2 / **31.5** |

attention, we observe consistent and substantial improvements for both R@K and mR@K on both the `base` and `novel` split. These results demonstrate the efficacy of our adapter and the necessity of incorporating cross-attention for capturing the complicated interplay between subjects and objects. Since DM is designed for geometric relations with massive annotations [21, 24], the improvements on R@K are considerable, while those on mR@K are relatively small. See §A for more results.

## 5 Conclusion

This work presents SDSGG, a *scene-specific* description based framework for OVSGG. Despite the previous works based upon category-/part-level prompts, we argue that these text classifiers are

*scene-agnostic*, which cannot adapt to specific contexts and may even be misleading. To address this, we carefully design a multi-persona collaboration strategy for generating flexible, context-aware SSDs, a *self-normalized* similarity computation module for renormalizing the influence of each SSD, and a mutual visual adapter that consists of several trainable lightweight modules for learning interaction-aware space. Our approach distinguishes itself by using SSDs derived from LLMs, which are tailored to the content of the presented image. This is further enhanced by MVA, which captures the underlying interaction patterns based on the semantic space of VLMs. We expect the introduction of SDSGG will not only set a new benchmark for OVSGG, but also encourage the community to explore the potential of integrating LLMs with VLMs for deeper, contextual understanding of images.

**Acknowledgement.** This work was supported by the National Science and Technology Major Project (No. 2023ZD0121300), the National Natural Science Foundation of China (No. 62372405), the Fundamental Research Funds for the Central Universities 226-2024-00058, National Key Laboratory of Human-Machine Hybrid Augmented Intelligence, Xi'an Jiaotong University (No. HMHAI-202403), Bytedance Doubao Fund, and the Earth System Big Data Platform of the School of Earth Sciences, Zhejiang University.

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

## Summary of the Appendix

For a better understanding of the main paper, we provide additional details in this supplementary material, which is organized as follows:

- §A introduces more ablative experiments.
- §B details the implementation details.
- §C provides the pseudo code of SDSGG.
- §D shows the generated *scene-specific* descriptions and the corresponding prompts.
- §E offers more qualitative results.
- §F discusses our limitations, societal impact, and directions of future work.
- §G presents more experimental results.

## A   More Ablative Experiment

**Self-normalized Similarity.** Furthermore, we study the effectiveness of *self-normalized* similarity for evaluating the influence of each *scene-specific* description (§3.1) in Table S1. Here we remove the opposite description and use only the original scene description as the baseline. As seen, without the reference of opposite description, the performance drops drastically, *e.g.*, 28.3%/25.5%/28.4% *vs*. **31.6%/30.0%/35.1%** R@100 & 12.1%/29.8%/24.7% *vs*. **14.7%/31.5%/28.6%** mR@100 on the `base`/`novel`/`semantic` split, respectively. This indicates the importance of renormalizing similarity scores, and also validates our hypothesis – the *absolute* value of one single similarity yields only limited insight, while the *difference* of two similarities provides more information.

Table S1: Ablation studies (§A) on *self-normalized similarity* (SNS, §3.1).

| Method | Split | R@20↑ | R@50↑ | R@100↑ | mR@20↑ | mR@50↑ | mR@100↑ |
|---|---|---|---|---|---|---|---|
| **Ours** | base | **18.7** | **26.6** | **31.6** | **9.1** | **12.3** | **14.7** |
| *w/o* SNS | | 16.1 | 23.3 | 28.3 | 7.0 | 10.0 | 12.1 |
| **Ours** | novel | **18.9** | **25.8** | **30.0** | **16.6** | **25.2** | **31.5** |
| *w/o* SNS | | 14.7 | 20.6 | 25.5 | 14.2 | 21.5 | 29.8 |

**Number of SSDs.** We first study the impact of the number of used classifiers (*i.e.*, SSDs). The results, presented in Table S2, reveal a considerable performance enhancement when the number of pairs increased from 11 to 21, affirming the efficiency of SSDs. However, we also witnessed a dip in performance when too many SSDs were employed. This could be due to the unnecessary pairing of a relation category with an excessive number of descriptions (52 descriptions for 26 pairs).

Table S2: Ablation studies (§A) on the number of used classifiers (Eq. 2). The adopted hyperparameters are marked in red.

| # Pairs | Split | R@20 | R@50 | R@100 | mR@20 | mR@50 | mR@100 |
|---|---|---|---|---|---|---|---|
| 11 | base | 13.9 | 19.8 | 24.0 | 8.6 | 12.0 | 14.3 |
| 16 | | 19.4 | 27.5 | 32.9 | 8.6 | 11.8 | 14.2 |
| 21 | | 18.7 | 26.6 | 31.6 | 9.1 | 12.3 | 14.7 |
| 26 | | 11.7 | 16.7 | 20.0 | 7.7 | 11.1 | 13.5 |
| 11 | novel | 18.4 | 23.9 | 26.8 | 16.6 | 23.3 | 27.2 |
| 16 | | 17.9 | 24.4 | 28.7 | 16.4 | 23.2 | 28.7 |
| 21 | | 18.9 | 25.8 | 30.0 | 16.6 | 25.2 | 31.5 |
| 26 | | 12.4 | 18.5 | 23.7 | 14.3 | 21.8 | 27.2 |

**Scaling Factors.** We then study the effectiveness of the scaling factors used in our training objectives (§3.3) in Table S3. As seen, the performance remains consistent for those compared hyperparameters. This indicates the robustness of SDSGG to changes in the scale of the training targets. Considering the performance on all metrics together, $\alpha$ and $\beta$ are set to be 2 and 1e-1, respectively.

**Number of Attention Heads.** We further investigate the impact of the number of attention heads $H$ used in the mutual visual adapter (§3.2, Eq. 5). As shown in Table S4, the performance improves

Table S3: Ablation studies (§A) on the scaling factors used in the training objectives, *i.e.*, $\alpha$ and $\beta$ (§3.3, Eq. 8). The adopted hyperparameters are marked in red.

| $\alpha$ | $\beta$ | Split | R@20↑ | R@50↑ | R@100↑ | mR@20↑ | mR@50↑ | mR@100↑ |
|---|---|---|---|---|---|---|---|---|
| 1.5 | 1e-1 | | 16.8 | 24.0 | 28.7 | 9.8 | 12.7 | 15.0 |
| 2.5 | 1e-1 | | 19.4 | 27.3 | 32.4 | 9.4 | 12.6 | 14.9 |
| 2 | 3e-1 | base | 18.8 | 26.7 | 31.8 | 9.4 | 12.6 | 14.9 |
| 2 | 5e-1 | | 17.2 | 24.4 | 29.1 | 9.2 | 12.2 | 14.4 |
| 2 | 0 | | 18.6 | 26.4 | 31.7 | 9.2 | 12.1 | 14.3 |
| 2 | 1e-1 | | 18.7 | 26.6 | 31.6 | 9.1 | 12.3 | 14.7 |
| 1.5 | 1e-1 | | 19.8 | 25.5 | 29.3 | 17.5 | 24.5 | 30.8 |
| 2.5 | 1e-1 | | 18.8 | 25.2 | 29.8 | 19.5 | 25.9 | 32.1 |
| 2 | 3e-1 | novel | 18.4 | 25.2 | 29.7 | 16.5 | 24.4 | 30.4 |
| 2 | 5e-1 | | 18.5 | 25.0 | 29.3 | 15.8 | 23.1 | 29.2 |
| 2 | 0 | | 18.8 | 25.5 | 29.6 | 17.3 | 23.6 | 30.1 |
| 2 | 1e-1 | | 18.9 | 25.8 | 30.0 | 16.6 | 25.2 | 31.5 |

from 32.7% to **35.1**% R@100 on the `semantic` split when increasing the number of heads from 4 to 8, but the number of parameters steadily increase as the number of heads grows. When increasing the number of heads from 8 to 16, we observe some improvements (*e.g.*, 31.6% to 32.8% R@100 on the `base` split), but also some performance drops (*e.g.*, 30.0% to 29.0% R@100 on the `novel` split). Consequently, we set $H = 8$ as the default to strike an optimal balance between accuracy and computation cost.

Table S4: Ablation studies (§A) on the number of attention heads $H$ used in the mutual visual adapter (§3.2, Eq. 5). The adopted hyperparameters are marked in red.

| $H$ | Param. | Split | R@20↑ | R@50↑ | R@100↑ | mR@20↑ | mR@50↑ | mR@100↑ |
|---|---|---|---|---|---|---|---|---|
| 4 | 4.2M | | 18.9 | 27.0 | 32.5 | 9.1 | 12.2 | 14.4 |
| 8 | 7.1M | base | 18.7 | 26.6 | 31.6 | 9.1 | 12.3 | 14.7 |
| 16 | 12.8M | | 19.4 | 27.5 | 32.8 | 9.5 | 12.7 | 15.1 |
| 4 | 4.2M | | 18.2 | 25.3 | 29.9 | 15.9 | 24.8 | 31.0 |
| 8 | 7.1M | novel | 18.9 | 25.8 | 30.0 | 16.6 | 25.2 | 31.5 |
| 16 | 12.8M | | 18.0 | 25.3 | 29.0 | 15.8 | 24.0 | 29.1 |

**Margin.** Last, we examine the impact of the margin $\lambda$ in our training objectives (§3.3, Eq. 8). As shown in Table S5, the performance remains consistent for those compared values, which indicates the robustness of our learning procedure. $\lambda$ is set to be 3e-2 by default.

Table S5: Ablation studies (§A) on the margin $\lambda$ (Eq. 8). The adopted hyperparameters are marked in red.

| $\lambda$ | Split | R@20 | R@50 | R@100 | mR@20 | mR@50 | mR@100 |
|---|---|---|---|---|---|---|---|
| 5e-2 | | 19.2 | 26.9 | 32.0 | 9.5 | 12.6 | 15.0 |
| 3e-2 | base | 18.7 | 26.6 | 31.6 | 9.1 | 12.3 | 14.7 |
| 1e-2 | | 18.6 | 26.0 | 30.8 | 8.8 | 11.8 | 13.8 |
| 5e-2 | | 18.7 | 24.8 | 29.1 | 17.0 | 24.0 | 30.3 |
| 3e-2 | novel | 18.9 | 25.8 | 30.0 | 16.6 | 25.2 | 31.5 |
| 1e-2 | | 18.0 | 24.6 | 28.7 | 16.6 | 24.4 | 29.2 |

# B  Implementation Details

**Training.** Our model is trained with a batch size of 4. One RTX 3090 is used for training. During the training process, images are resized to dimensions within the range of [600, 1,000]. For each relation, up to 50K samples are included. Random flipping is adopted for data augmentation. SGD is adopted for optimization. The initial learning rate, momentum, and weight decay are set to be 2e-2, 9e-1, 1e-4, respectively. We utilize the pre-trained weights of CLIP to initialize our model. To avoid data leakage, we remove annotations in the training set which contains categories in the `novel` split.

**Testing.** Testing is conducted on the same machine as in training. No data augmentation is used during testing. In terms of the `semantic` split, we directly applied the same weights trained on the `base` split for testing. A similar filtering strategy [12] is adopted.

**Codebase and Architecture.** We use the same codebase as in [21]. We adopt GPT-3.5 from OpenAI as our LLM. In terms of CLIP, we employ a widely used architecture, *i.e.*, ViT-B/32, for initializing our mutual visual adapter.

**Split.** We visualize all relation categories in Table S6. To guarantee fairness, we select GQA's [15] relation categories that also exist in VG [14]. As such, we can reuse SSDs obtained in our experiments on VG, thus verifying the generalizability of the proposed SSDs and self-normalizing similarity.

Table S6: Relation categories in each split.

| Split | Relation Categories |
|---|---|
| Base | watching, of, hanging from, to, near, carrying, parked on, covered in, wearing, sitting on, made of, on, standing on, from, in front of, belonging to, between, above, attached to, walking on, behind, in, holding, against, has, looking at, under, at, playing, riding, covering, for, with, wears, over |
| Novel | flying in, painted on, mounted on, using, and, on back of, growing on, lying on, along, part of, eating, laying on, walking in, across, says |
| Semantic | watching, growing on, hanging from, eating, carrying, parked on, covered in, says, using, flying in, painted on, sitting on, lying on, standing on, walking in, mounted on, attached to, walking on, holding, looking at, playing, riding, covering, laying on |

(a) VG [14]

| Split | Relation Categories |
|---|---|
| Base | in front of, watching, riding, covered in, under, wearing, behind, parked on, covering, above, of, on, holding, walking on, at, carrying, with, sitting on, in, standing on, near |
| Novel | lying on, looking at, growing on, walking in, attached to, using, mounted on, hanging from, flying in, eating |

(b) GQA [15]

## C  Pseudo Code

The pseudo code of SDSGG is given in Algorithm S1 and Algorithm S2. We respectfully refer the reviewer to the supplementary Python files for the PyTorch implementation of SDSGG's key modules. Moreover, to guarantee reproducibility, our full code and pre-trained models will be publicly available.

---

**Algorithm S1** Pseudo-code for MVA of SDSGG in a PyTorch-like style.

---

```
"""
img_feats1: subject or object visual features, where the first vector is CLS feature, and the left
    vectors are patch features.
img_feats2: subject or object visual features. The format is the same as img_feats1.
dire_mark_feature: marker used to identify the direction of the relation.
"""
def MVA(img_feats1, img_feats2, dire_mark_feat):
    # Project features into low-dimensional, interation-aware space.
    patch_feats = LinearDown(img_feats1[1:]) # Eq. 4

    # Model the interplay between the subject and object
    out = CrossAttn(patch_feats, img_feats2[1:]) # Eq. 5

    # Project features into the original dimension.
    out = LinearUp(out) # Eq. 5

    # Add direction marker.
    mva_out = Linear(Cat([out, dire_mark_feature]))
    mva_out = (mva_out + img_feats1[0]).mean() # Eq. 6
    return mva_out
```

---

**Algorithm S2** Pseudo-code for the forward process of SDSGG in a PyTorch-like style.

```
"""
bboxes: bounding boxes of all objects.
img: input image.
text_raw_des: raw description.
text_op_des: opposite description.
asso: the association presented in "Association between Scene-level Descriptions and Relation Categories".
targets: training targets.
"""

def forward(bboxes, img, text_raw_des, text_op_des, asso, targets):

    # Batch crop and encode images.
    crop_img = Crop(img, bboxes)
    img_feats = EncodeImage(crop_img) # CLIP's visual encoder, Eq. 3

    # Tokenize and encode descriptive text.
    text_raw_feats = EncodeText(Tokenize(text_raw_des)) # CLIP's text encoder
    text_op_feats = EncodeText(Tokenize(text_op_des))

    # Define directional marker
    mark_sub = EncodeText(Tokenize("a photo of subject"))
    mark_obj = EncodeText(Tokenize("a photo of object"))

    scores, losses = []
    # Calculate self-normalized similarity for each subject-object pair
    for sub, obj in sub_obj_pairs:

        mva_out_s2o = MVA(img_feats[sub], img_feats[obj], mark_sub)
        mva_out_o2s = MVA(img_feats[obj], img_feats[sub], mark_obj)
        mva_out = (mva_out_s2o + mva_out_o2s) / 2.0

        sim_raw = CosSim(mva_out, text_raw_feats)
        sim_op = CosSim(mva_out, text_op_feats)

        score = ((sim_raw - sim_op) * asso).sum() # Eq. 2
        scores.append(score)

        # If is training
        if training:
            loss = compute_loss(sim_raw, sim_op, targets) # Eq. 8
            losses.append(loss)

    return scores, losses
```

# D    Scene-specific Description

The detailed, *scene-specific* descriptions generated by LLM's multi-persona collaboration are provided in Fig. S1. As seen, our *scene-specific* descriptions (21 pairs, 42 descriptions in total) cover different scenes from a comprehensive view, *e.g.*, *spatial* descriptions ("two or more objects partially overlap each other", "interaction between objects", *etc.*), *environment* (background) descriptions ("on a road", "on a flat plane, it should appear balanced with no visible tilting", *etc.*), *semantic* descriptions ("belong to animal or human behavior", "may have contact behavior", *etc.*), *appearance* descriptions ("might have flat teeth or sharp teeth", "have a curvy body", *etc.*).

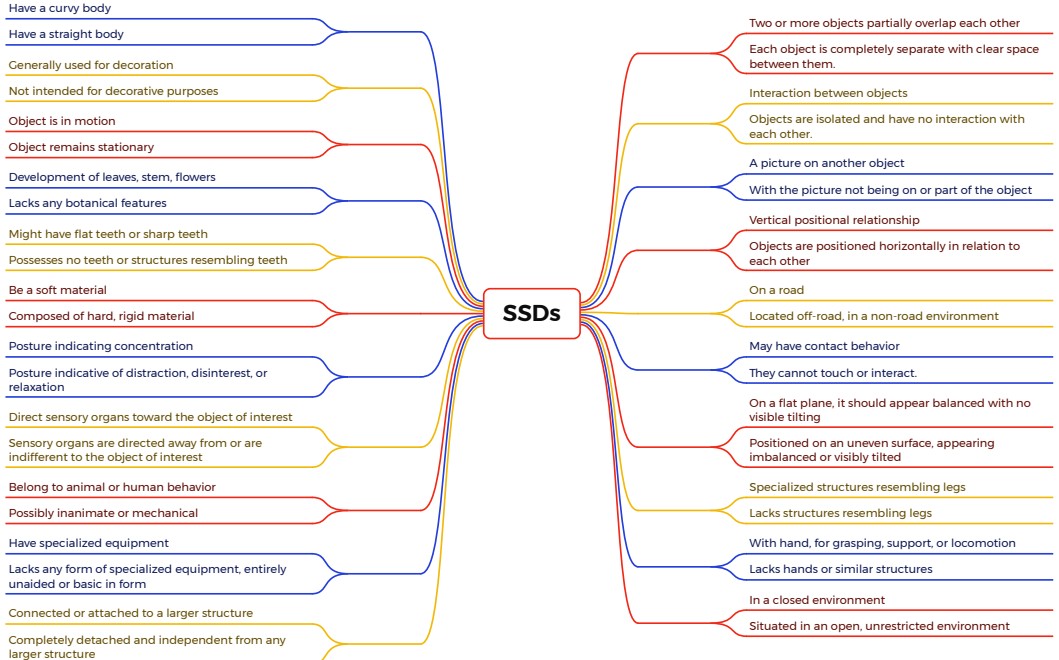

Figure S1: Illustration of the generated scene-specific descriptions.

Next, we detail each step to generate scene-specific descriptions.

❶ **Initial Description Generation.** This step can be repeated many times to generate a large number (72 after manual selection) of scene descriptions. The prompt is shown in Fig. S2.

❷ **Summarizing Descriptions.** Since these initial descriptions may suffer from noise and semantic overlap, we ask LLM to streamline and combine these descriptions, ensuring more cohesive and distinct scene-level descriptions. The prompt is shown in Fig. S3.

❸ **Description-Relation Association.** After obtaining various scene descriptions, a critical inquiry arises regarding their utility for relation detection, given their lack of explicit association with specific relation categories. To address this, we delineate three distinct scenarios characterizing the interplay between relation categories and scene descriptions: **i**) certain coexistence ($C_r^n = 1$), where a direct correlation exists; **ii**) possible coexistence ($C_r^n = 0$), indicating a potential but not guaranteed association; and **iii**) contradiction ($C_r^n = -1$), denoting an incompatibility between the scene description and relation category. Here $C_r^n$ denotes the correlation between relation $r \in \mathcal{R}$ and $n_{th}$ scene description, and is generated by LLMs. The prompt is shown in Fig. S4.

❹ **Opposite Description Generation.** Since classifiers are contextually bound, we generate opposite descriptions to compute *self-normalized* similarities. The prompt is shown in Fig. S5.

```
# Input: {scene content to be discussed}
# Output: 3~5 descriptions
"""
Begin by embodying three distinct personas: a biology expert, a physics expert, and
    an engineering expert. Each expert will articulate a step-by-step approach
    along with their thought process, considering various hypothetical scenarios
    relevant to their field of study, and then share their insights with the
    group. If any expert does not know the answer, he will exit the discussion.
    Once all experts have provided their analyses, summarize the final generic
    scene descriptions.
The generic scene description involves the specific appearance and the visible
    action/motion/interaction may appear in the scene (use your imagination here).
    Here are some examples of generic scene descriptions:
{some in-context examples}
Here is an example of discussion:
Discussion started!
Question: Suppose there is ... (this involves a detailed discussion of three roles)

Discussion started!
Question: Suppose there is {scene content to be discussed}, please give concise,
    generic scene descriptions of this scene.
"""
```

Figure S2: Prompts for initial description generation.

```
# Input: initial {scene descriptions}, all {relation categories}, {number of final
    scene descriptions}
# Output: final scene-level descriptions
"""
Here is a text pool that includes a series of descriptive texts:
{scene descriptions}
Here are all the relation categories:
{relation categories}
You are asked to pick {number of final scene descriptions} descriptive statements
    from the text pool that can describe at least two predicates. Think step by
    step.
"""
```

Figure S3: Prompts for summarizing descriptions.

```
# Input: {scene descriptions}, all {relation categories}
# Output: description-relation associations
"""
For {scene descriptions}, decide whether it is likely to appear in one of the
    following flat photos where {relation categories} appears.
The judgment result is to choose between [certain coexistence, possible
    coexistence, and contradiction]. When the photo shows that the scene must have
    a certain description, it is judged as "contradiction". When the photo has
    little relationship with the description, it is judged as "possible
    coexistence". When the photo shows that the scene must not appear with a
    certain description, it is judged as "contradiction".
"""
```

Figure S4: Prompts for generating description-relation associations.

```
# Input: {scene descriptions} (original scene descriptions)
# Output: final scene-level descriptions
"""
Here are some descriptions:
{scene descriptions}
Please give their opposite descriptions.
"""
```

Figure S5: Prompts for opposite description generation.

# E   More Qualitative Comparison Result

We provide more visual results that compare SDSGG to CLIP [4] in Fig. S6. It can be observed that SDSGG performs robust in hard cases and can consistently deliver more satisfying results, based upon the *scene-specific* descriptions and *self-normalized* similarities.

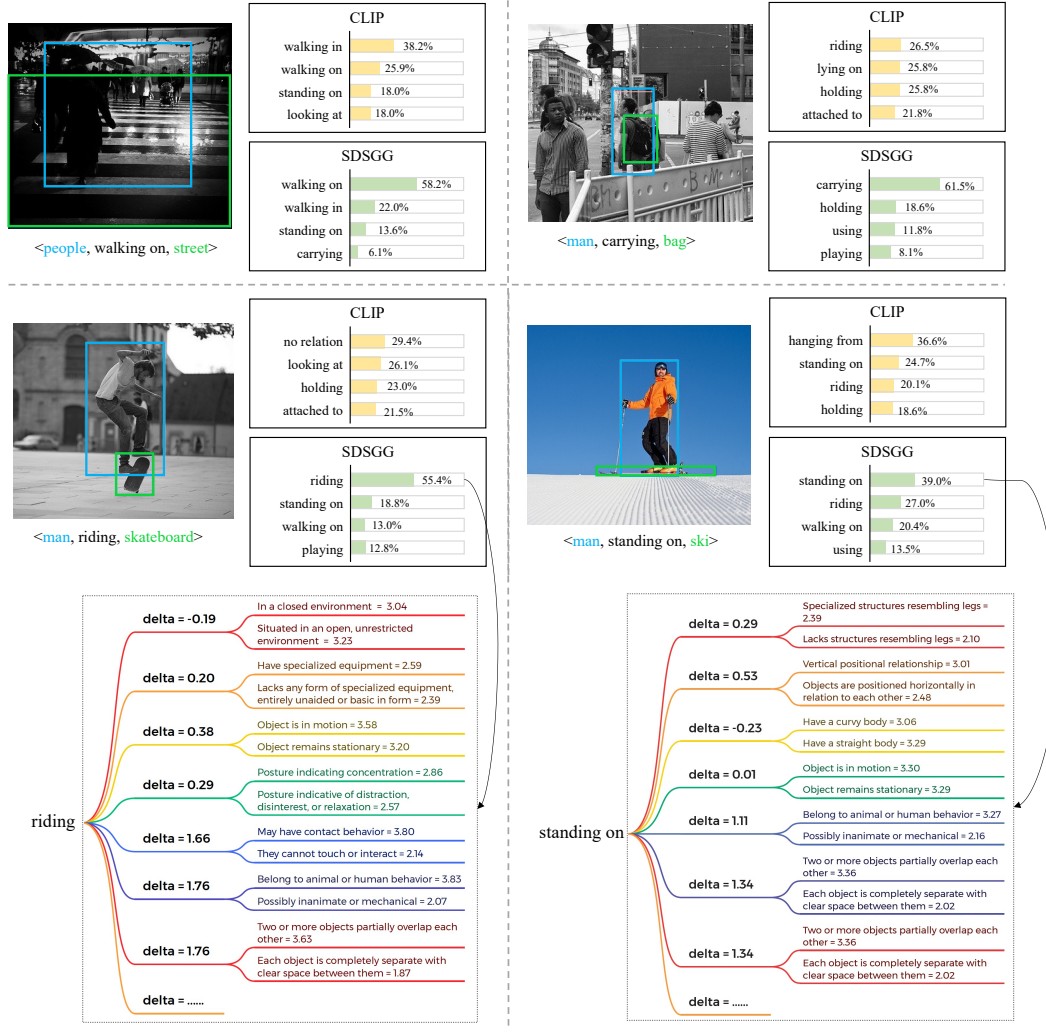

Figure S6: Visual results (§E) on VG [14] `test`. As seen, SDSGG makes the right predictions based upon the *scene-specific* descriptions and *self-normalized* similarities.

# F  Discussion

**Limitation Analysis.** Following [12], currently our algorithm is specifically designed for the predicate classification task [1, 16] to make fair comparisons with relation detection models. Such a pairwise classification task gives the ground-truth annotations of objects in the scene for easier representation learning of relations. However, these annotations are unavailable in real-world applications.

**Societal Impact.** This work points out the drawbacks of existing *scene-agnostic* text classifiers and accordingly introduces several new modules for both the visual and textual components, leading to a *scene-specific* description based OVSGG framework that combines the strengths of VLMs and LLMs. Like every coin has two sides, using our framework will have both positive and negative impacts. On the positive side, our work contributes to research on intelligent scene understanding, and thus is expected to eventually benefit industries such as autonomous driving. For potential negative social impact, the reliance on VLMs and LLMs could lead to the perpetuation of biases and inequalities present in the data used in these models' large-scale pre-training stage.

**Future Work.** As mentioned before, the focus of this work is not on object detection. It is interesting to extend our algorithm to handle the object detection task simultaneously by, for example, incorporating set-prediction architectures [91]. Moreover, the design of our multi-persona collaboration stands for an early attempt and deserves to be further explored. In addition, the architectural designs of directional marker and mutual visual adapter certainly worth further explorations, *e.g.*, efficiency [92], architecture [93, 94], and adaptive prompting [95, 96]. Furthermore, extending our algorithm to other relation detection tasks [97, 40, 42, 98] may lead to an uniformed relation detection algorithm.

# G  More Experiment

**Training on the Full Set of Relation.** We trained our model with frequency bias [25] on the full set of relations. The results are shown in Table S7.

Table S7: Results on the full set of relation.

| Method | mR@50↑ | mR@100↑ |
|--------|--------|---------|
| Ours | 28.7 | 34.2 |

**Different Base/Novel Splits.** We trained our model on different base/novel splits to investigate the robustness further. Specifically, we **i**) change the proportion of the base and novel split and **ii**) change the categories within the base and novel split (*i.e.*, different No. for the same ratio). The results are shown in Table S8.

Table S8: Results on different base/novel splits.

| No. | base:novel | Base | | Novel | |
|-----|-----------|------|------|-------|------|
| | | mR@50↑ | mR@100↑ | mR@50↑ | mR@100↑ |
| 1 (paper) | 35:15 | 12.3 | 14.7 | 25.2 | 31.5 |
| 2 | 35:15 | 12.4 | 14.8 | 24.3 | 28.2 |
| 3 | 32:18 | 11.9 | 14.4 | 23.9 | 28.4 |
| 4 | 32:18 | 13.6 | 15.9 | 20.6 | 26.7 |
| 5 | 38:12 | 11.8 | 14.2 | 23.7 | 29.9 |
| 6 | 38:12 | 11.5 | 13.7 | 22.6 | 27.1 |

**Inference Time.** Since the renormalization and similarity measurement involve only a few matrix operations that can be omitted from the complexity analysis, we will focus on the inference time of three main modules. The results are shown in Table S9.

Table S9: Inference time for different modules.

| Module | Inference Time (ms/pair) |
|--------|--------------------------|
| CLIP's Visual Encoder | 6.5 |
| Mutual Visual Adapter | 0.2 |
| CLIP's Text Encoder | 5.4 |

**Different Personas.** The involvement of multiple personas in the discussion process enhances the diversity of generated descriptions. The key point of our multi-persona collaboration is about the "collaboration" rather than a specific persona. Actually, using only one persona can even decrease the diversity of generated descriptions and hurt the performance, as it can only generate descriptions from its own viewpoint without discussion with others. In addition, we changed the system prompt of LLM from the default like "you are a helpful AI assistant" into a persona-specific one like "you are a biologist". We then evaluate the performance of our model with these generated descriptions. The results are shown in Table S10.

Table S10: Comparison of different personas.

| Method | Base | | Novel | |
|---|---|---|---|---|
| | mR@50↑ | mR@100↑ | mR@50↑ | mR@100↑ |
| Multi-persona Collaboration | 12.3 | 14.7 | 25.2 | 31.5 |
| Biologist Persona | 6.3 | 8.4 | 12.9 | 18.4 |
| Engineer Persona | 7.3 | 9.7 | 8.4 | 11.5 |
| Physicist Persona | 4.8 | 6.9 | 9.1 | 14.7 |

