# OpenReview forum: "Scene Graph Generation with Role-Playing Large Language Models"
_NeurIPS.cc/2024/Conference — NeurIPS 2024 poster_

### Official Review · Reviewer_xMLQ · 2024-07-08

**Soundness:** 3
**Presentation:** 3
**Contribution:** 3
**Rating:** 6
**Confidence:** 3

**Summary:**

This paper proposes SDSGG, a novel open vocabulary scene graph generation(OVSGG) algorithm that leverages the reasoning capability of a LLM to better determine the relations between objects in the scene. It achieves this goal by first prompting a LLM with multiple persona prompts to expand a simple relational predicative to a list of detailed visual descriptions, which are subsequently used to augment the classification process.  It also introduces a novel mutual visual adapter, which better captures the interaction between subjects and objects.  Experiments show that these proposed designs are effective.

**Strengths:**

1. Incorporating a LLM to augment the predicate labels for scene graph generation is a novel idea.  This paper provides meaningful insight to future works in this area.
2. The experiment results (table 1-2) are strong, significantly outperforming previous methods.
3. The authors conducted extensive ablation studies on various design elements.

**Weaknesses:**

1. Prompting the LLM is a key element of the method, however some crucial details are missing. For example, how are the prompts constructed? While the author provided the prompt in Appendix Fig 5, it is unclear how the  "{scene content to be discussed}" is generated. The author did show some examples throughout the paper, but they are not sufficient for the reader to understand the underlying process. In particular, in L167, the author showed example #1 "Imagine there is an animal that is eating, ". In Fig 1c, there is example #2 "Assuming that the scene has a man riding a horse."  These two descriptions have two different granularity, as one only includes the generic concept of "an animal that is eating" while the other has specific class names "man" and "horse". The authors should clearly describe what information is included into the prompt, and discuss the scalability and cost of generating such prompts. I suppose if the prompts are like example #1, they can be generated offline based on predicative label sets. However, if the prompts are like example #2, they need to be generated for every possible triple of (subject, predicative, object) over the label space, or be generated online over possible objects in a scene. It is unclear which is the case.

2. Additional discussions and experiments are required to justify some of the design choices. For example,

  2.1 in eq 8, the loss of descriptions marked by possible coexistence is to make the prediction "close to those of CLIP." (L255). If this is the case, why not directly use CLIP results for these possible coexistence descriptions at inference time (eq 2)?

  2.2 some discussion is needed on if CLIP is good at classifying the generated descriptions. What are the nature of these descriptions and do they fit well with CLIP's pretraining pipeline (i.e. object-level image caption)? As a concrete example, can CLIP properly distinguish descriptions involving counting, such as "with four legs", and "with two legs", mentioned in the examples?

 2.3 what happens if we discard "possible coexistence" descriptions and only use definite coexistence and contradiction? Table8 shows that it is ideal to have a low weight for "possible coexistence"  loss. What happens if we set the weight to 0 and remove it at inference pipeline?

**Questions:**

See weakness.

**Limitations:**

The authors discussed limitations and societal impact.

---

> ### Author Rebuttal · Authors · 2024-08-06
>
> We thank reviewer xMLQ for the valuable time and constructive feedback. We provide point-to-point response below.
>
> **Q1**: **Clarifying the rule of prompt construction.**
>
> **A1**: The prompts **are like example #1** and **are generated offline**. The "{scene content to be discussed}" is constructed based on a given *subject-predicate* or *predicate-object* pair. For instance, in L167 (example #1), the "{scene content to be discussed}" is constructed based on a *subject-predicate* pair where the subject is “animal” and the predicate is “eating”. Details about the scalability and cost of generating descriptions are provided in Q2.
>
> Regarding Fig. 1c, it is indeed misleading. In Fig. 1c, we specify a triplet instead of a pair for easier understanding of the later discussion process.  We appreciate the keen observations and recognize that this creates an inconsistency. We will revise Fig. 1c for clarity. The revised version is shown in the PDF (Fig._R1). Thank you.
>
> **Q2**: **Scalability and cost of generating descriptions.**
>
> **A2**: As you noticed, iterating all possible triples of <subject, predicate, object> is **computationally unaffordable** (150 * 50 * 150 for 150 object categories and 50 predicate categories). Therefore, we use *subject-predicate* and *predicate-object* pairs to generate descriptions. Following [a], we use 3 object categories (*i.e*., human, animal, and product) instead of 150 object categories to save computational costs. The description generation process (as mentioned in L653-667) involves the following steps:
>
> 1. Initial Description Generation. This step calls OpenAI's API 2 * 3 * 50 times (2 stands for *subject-predicate* and *predicate-object* pairs).
> 2. Summarizing Descriptions. This step calls OpenAI's API 1 time.
> 3. Description-Relation Association. This step calls OpenAI's API 50 times.
> 4. Opposite Description Generation. This step calls OpenAI's API 1 time.
>
> As illustrated, the primary cost is in the initial description generation (i.e., discussion by multi-persona collaboration). The generation is one-time and is completed **offline**, and would not delay the inference stage. Since NOT all triples are iterated, the cost remains acceptable even when the number of object/predicate categories increases. We respectfully refer the reviewer to {reviewer RK4c Q2} for the cost of online deployment.
>
> **Q3**: **Usage of possible coexistence descriptions.**
>
> **A3**: The goal of making the prediction of possible coexistence descriptions close to those of CLIP is to regularize the training process and avoid overfitting (L255-257). Such an approach brings more supervision signals during training and is inspired by the knowledge distillation-based open-vocabulary frameworks [b]. It is important to note that possible coexistence descriptions are used only during training. **During inference, they DO NOT contribute to the similarity measurement**, as the correlation $C_{r}^{n}$ equals 0 (see Eq. 1 & 2).
>
> **Q4**: **More ablative experiments on possible coexistence descriptions.**
>
> **A4**: In Table 8, $\alpha_{indef}$ is a scaling factor (Eq. 8, L258) for training targets rather than a weight of a loss. Per your request, we remove $\mathcal{L}_{indef}$ in Eq. 9 to study its influence. The results are as follows:
>
> | Method | Base |  | Novel |  |
> | --- | --- | --- | --- | --- |
> |  | mR @50 | mR @100 | mR @50 | mR @100 |
> | Ours | 12.3 | 14.7 | 25.2 | 31.5 |
> | w/o $\mathcal{L}_{indef}$ | 12.1 | 14.3 | 23.6 | 30.1 |
>
> As seen, the $\mathcal{L}_{indef}$ leads to considerable improvements in the novel set (e.g., 1.6% mR@50 and 1.4% mR@100) and does not affect the performance of the base set. This further varifies our hypothesis (*i.e.*, regularize the training process and avoid overfitting as stated in L255-257). We will incorporate the above results into the appendix of our revision. Thanks.
>
> **Q5**: **Discussion on descriptions and CLIP's recognition capability.**
>
> **A5**: The generated description can be viewed as a **semantically informative** representation compared to the raw representation with only category names. They can convey extra concepts *without modifying model architecture or introducing additional learning processes*. Concretely, they 1) provide comprehensive concepts that help CLIP differentiate between relations (see Fig. 1 in [c]), and 2) offer simpler and more common concepts for relation categories that may not have been encountered during CLIP's pre-training (see Fig. 5 in [d]).
>
> Note that these detailed, fine-grained descriptions **may not align perfectly with CLIP's pre-training pipeline**. By collecting 10 horse images from the Web and comparing the vision-language similarities, we empirically find that CLIP struggles to properly distinguish between concepts with slight differences (*e.g.*, "with four legs" *vs*. "with two legs"). This indicates an inherent drawback of previous description-based methods that uniformly process all descriptions as affirmative classifiers (L50), and motivates us to introduce the renormalization mechanism (L69-76) so as to adaptively weight the generated descriptions. We will incorporate the discussions into the appendix of our revision. Thank you.
>
> References:
>
> [a] Unbiased Heterogeneous Scene Graph Generation with Relation-aware Message Passing Neural Network, AAAI 2023.
>
> [b] Towards Open Vocabulary Learning: A Survey, TPAMI 2024.
>
> [c] Zero-shot Visual Relation Detection via Composite Visual Cues from Large Language Models, NeurIPS 2023.
>
> [d] Visual Classification via Description from Large Language Models, ICLR 2022.

---

> > ### Comment · Reviewer_xMLQ · 2024-08-12
> >
> > The authors successfully addressed my concerns.  I especially appreciate the clarifications of Q1. I maintain my recommendation for acceptance.

---

> > > ### Author Response · Authors · 2024-08-13
> > > **Thanks for your review**
> > >
> > > Dear Reviewer xMLQ,
> > >
> > > We sincerely appreciate your time and effort in reviewing our submission and providing valuable comments. Please let us know if you'd like any further information.
> > >
> > > Sincerely yours,
> > >
> > > Authors.

---

### Official Review · Reviewer_RK4c · 2024-07-10

**Soundness:** 3
**Presentation:** 3
**Contribution:** 3
**Rating:** 6
**Confidence:** 4

**Summary:**

This paper aims to solve the open-vocabulary scene graph generation problem. Previous methods mainly adopt scene-agnostic prompts as text classifiers. The authors argue that using the fixed text classifiers not only struggles to model visual relations with high variance, but also falls short in adapting to distinct contexts. Therefore, the authors propose the scene-specific description based OVSGG framework. They employ an LLM and ask it to play different roles. Besides, they design the mutual visual adapter to encode visual features. Extensive experiments show that the proposed method significantly outperforms top-leading methods.

**Strengths:**

The motivation and idea of this paper are innovative and interesting. Simply applying LLM to SGG cannot effectively reason the relationships. The authors consider employing the context and introducing multiple roles of LLM, which is shown to be effective for solving the OVSGG problem.

Besides, the experiments are convincing. Plenty of ablation studies are provided.

**Weaknesses:**

My main concern is Computational Complexity: The proposed framework involves multiple stages, including generating descriptions, renormalizing them, and applying mutual visual adapters. This multi-step process could be computationally intensive, making it less practical for real-time applications or scenarios with limited computational resources.

**Questions:**

Please read the weaknesses part.

**Limitations:**

Yes

---

> ### Author Rebuttal · Authors · 2024-08-06
>
> We thank reviewer RK4c for the valuable time and constructive feedback. We provide point-to-point response below.
>
> **Q1**: **Computational complexity of description generation (offline).**
>
> **A1**: Suppose there are 3 common object categories (*i.e.*, human, animal, and product [a,b]) and 50 predicate categories. The description generation process (L653-667) involves the following steps:
>
> 1. Initial Description Generation. This step calls OpenAI's API 2 * 3 * 50 times.
> 2. Summarizing Descriptions. This step calls OpenAI's API 1 time.
> 3. Description-Relation Association. This step calls OpenAI's API 50 times.
> 4. Opposite Description Generation. This step calls OpenAI's API 1 time.
>
> The execution time of the above steps depends on the network and the speed of OpenAI's response. Since these descriptions are generated offline, **they DO NOT incur any computational cost during deployment.** We respectfully refer the reviewer to {reviewer xMLQ Q2} for the scalability and cost of generating descriptions.
>
> **Q2**: **Computational complexity of model (online).**
>
> **A2**: Since the renormalization and similarity measurement (Eq. 2) involve only a few matrix operations that can be omitted from the complexity analysis, we will focus on reporting the inference time of the following three main modules:
>
> | Module | Inference Time (ms) |
> | --- | --- |
> | CLIP’s Visual Encoder | 6.5 |
> | Mutual Visual Adapter | 0.2 |
> | CLIP’s Text Encoder | 5.4 |
>
> As seen, the inference time of CLIP’s visual and text encoder is significantly higher than that of our mutual visual adapter. Compared to [b] which also uses CLIP, the delay of the newly-introduced MVA (0.2ms) is neglectable. In addition, it is important to note that during deployment, **only the visual part** (*i.e.*, CLIP’s visual encoder and our mutual visual adapter) **requires computational resources.** This is because the descriptions are generated offline for all categories and remain unchanged during deployment so that their text embedding can be pre-computed and stored. Therefore, the overall framework can meet the requirement of real-time applications or resource-limited scenarios. We will incorporate the results about computational complexity into the appendix of the revision.
>
> Reference:
>
> [a] Unbiased Heterogeneous Scene Graph Generation with Relation-aware Message Passing Neural Network, AAAI 2023.
>
> [b] Zero-shot Visual Relation Detection via Composite Visual Cues from Large Language Models, NeurIPS 2023.

---

> > ### Comment · Reviewer_RK4c · 2024-08-12
> >
> > I am satisfying with the responses. I will keep my original rating.

---

> > > ### Author Response · Authors · 2024-08-13
> > > **Thanks for your review**
> > >
> > > Dear Reviewer RK4c,
> > >
> > > Thank you again for your kind review and comments. Please let us know if you'd like any further information.
> > >
> > > Sincerely yours,
> > >
> > > Authors.

---

### Official Review · Reviewer_qa1m · 2024-07-12

**Soundness:** 3
**Presentation:** 3
**Contribution:** 3
**Rating:** 6
**Confidence:** 5

**Summary:**

This paper starts by discussing methods for Open-vocabulary Scene Graph Generation (OVSGG) based on the CLIP model, highlighting the issue that current OVSGG methods do not differentiate between various scenes, which limits their effectiveness. The authors introduce SDSGG, a scene-specific description-based OVSGG framework that improves both the textual and visual parts, enhancing the model's open-vocabulary relationship prediction capabilities.

**Strengths:**

1. The novelty of this paper lies in its analysis of the issues present in current OVSGG methods, leading to the conclusion that differentiating between scenes is necessary to enhance the performance of OVSGG. The proposed Scene-specific Descriptions are particularly insightful.
2. The paper validates its findings on two datasets, VG and GQA, with experimental results showing significant performance improvements over previous state-of-the-art methods.

**Weaknesses:**

1. The description in Sec3.1, Scene-specific Text Classifiers, of the paper is somewhat confusing. This confusion arises primarily because the text section includes multiple different naming conventions and several distinct modules. It is recommended that this section be rewritten to make it easier for readers to understand. Additionally, the terminology used in this section is inconsistent with that in lines 64~77, leading to comprehension difficulties.
2. For the OVSGG method, it is suggested to also train the model on a full set of relations and compare its performance with conventional SGG methods to ensure that it achieves good performance under standard settings.
3. Is the model robust to different base/novel splits? It is recommended to train and test the model on different base/novel dataset divisions to assess its robustness.
4. It is advised to train and test the model on the PSG dataset as well.

**Questions:**

1. Regarding the selection of multiple personas, the ablation study shows that not using this approach results in a significant performance decrease. My question is, what exactly are the "standard prompts" referred to in line 329 of the document? What would be the effect if only one persona is used, and among the three personas mentioned in the document, which persona demonstrates the most significant performance?

---

> ### Author Rebuttal · Authors · 2024-08-07
>
> We thank reviewer qa1m for the valuable time and constructive feedback. We provide point-to-point response below.
>
> **Q1**: **Presentation.**
>
> **A1**: Our apologies. We will revise Section 3.1 to improve clarity and coherence. Our revisions will focus on:
>
> 1. Streamlining the naming conventions: We will eliminate the term "text classifiers" and consistently use "descriptions" throughout.
> 2. Delineating modules: We will add a summary of the newly-involved modules and their utility for description generation.
> 3. Ensuring consistency: We will standardize the terminology across the manuscript, aligning it with the definitions provided in L64~77.
>
> **Q2**: **Comparison with conventional SGG methods.**
>
> **A2**: Thanks for your suggestion. Per your request, we trained our model with frequency bias on the full set of relations. The results are as follows:
>
> | Method | mR@50 | mR@100 |
> | --- | --- | --- |
> | Motifs$_{CVPR'18}$ | 14.9 | 16.3 |
> | VCTree$_{CVPR'20}$ | 16.7 | 17.9 |
> | RU-Net$_{CVPR'22}$ | - | 24.2 |
> | PE-Net(P)$_{CVPR'23}$ | 23.1 | 25.4 |
> | VETO$_{ICCV'23}$ | 22.8 | 24.7 |
> | DRM$_{CVPR'24}$ | 23.3 | 25.6 |
> | Ours | 28.7 | 34.2 |
>
> As seen, **our model also demonstrates good performance under standard settings**. We will incorporate the above results into the appendix of our revision. Thanks.
>
> **Q3**: **Experiments on different base/novel splits.**
>
> **A3**: Good suggestion! As you noticed, the mean and variance are reported in Tables 1-3 to ensure the performance advantage is reliable. Per your request, we trained our model on different base/novel splits to investigate the robustness further. Specifically, we 1) change the proportion of the base and novel split and 2) change the categories within the base and novel split (*i.e.*, different No. for the same ratio).  The results are as follows:
>
> |  |  | Base |  | Novel |  |
> | --- | --- | --- | --- | --- | --- |
> | No. | base:novel | mR @50 | mR @100 | mR @50 | mR @100 |
> | 1 (paper) | 35:15 | 12.3 | 14.7 | 25.2 | 31.5 |
> | 2  | 35:15 | 12.4 | 14.8 | 24.3 | 28.2 |
> | 3 | 32:18 | 11.9 | 14.4 | 23.9 | 28.4 |
> | 4 | 32:18 | 13.6 | 15.9 | 20.6 | 26.7 |
> | 5 | 38:12 | 11.8 | 14.2 | 23.7 | 29.9 |
> | 6 | 38:12 | 11.5 | 13.7 | 22.6 | 27.1 |
>
> As seen, **our model is robust to different base/novel splits.** We will incorporate the above results into the appendix of our revision. Thank you.
>
> **Q4**: **Experiments on panoptic scene graph generation.**
>
> **A4**: Thanks for your suggestion. Our current framework is not directly applicable to PSG. Due to time constraints, we are unable to provide empirical results on the PSG dataset, as significant modifications and engineering efforts are needed. Nonetheless, we recognize the importance of exploring this direction and will definitely consider this as our future work. Thanks.
>
> **Q5: Standard prompts in L329.**
>
> **A5**: Given a *subject-predicate* or *predicate-object* pair, we ask LLM to generate corresponding descriptions. The prompt is defined as:
>
> > Imagine [*subject-predicate* / *predicate-object* pair]. Think about what the scene should look like. Summarize each descriptive statement of the scene in about 15 words each.
> >
>
> We respectfully refer the reviewer to {reviewer xMLQ Q1} for the rule of prompt construction.
>
> **Q6**: **Clarifying multi-persona collaboration.**
>
> **A6**: As mentioned in L336, the involvement of multiple personas in the **discussion process** enhances the diversity of generated descriptions. The key point of our multi-persona collaboration is about the “**collaboration**” rather than a specific persona. Actually, using only one persona can even decrease the diversity of generated descriptions and hurt the performance, as it can only generate descriptions from its own viewpoint without discussion with others. From the perspective of the number of generated descriptions, each persona's contribution is almost the same.
>
> In addition, we use the standard prompts and change the system prompt of LLM from the default like “you are a helpful AI assistant” into a persona-specific one like “you are a biologist”. We then evaluate the performance of our model with these generated descriptions. We only report the effect of the biologist and engineer persona because of the urgent due. The results are as follows:
>
> | Method | Base |  | Novel |  |
> | --- | --- | --- | --- | --- |
> |  | mR @50 | mR @100 | mR @50 | mR @100 |
> | Multi-pensona Collaboration | 12.3 | 14.7 | 25.2 | 31.5 |
> | Biologist Persona | 6.3 | 8.4 | 12.9 | 18.4 |
> | Engineer Persona | 7.3 | 9.7 | 8.4 | 11.5 |
> | Physicist Persona |  |  |  |  |
>
> As seen, using only one persona results in a significant performance decrease, which is consistent with the findings in Table 4. We will give the results of the physicist persona *asap*. Thank you.
>
> References:
>
> [a] Zero-shot Visual Relation Detection via Composite Visual Cues from Large Language Models, NeurIPS 2023.

---

> > ### Author Response · Authors · 2024-08-12
> >
> > Dear Reviewer qa1m,
> >
> > The results of the physicist persona are as follows:
> >
> > | Method | Base |  | Novel |  |
> > | --- | --- | --- | --- | --- |
> > |  | mR @50 | mR @100 | mR @50 | mR @100 |
> > | Physicist Persona | 4.8 | 6.9 | 9.1 | 14.7 |
> >
> > We appreciate the constructive feedback you provided, and we will incorporate all relevant results and details into our revision.
> >
> > Sincerely yours,
> >
> > Authors.

---

> > > ### Comment · Reviewer_qa1m · 2024-08-13
> > >
> > > I appreciate the effort you put into addressing my concerns in your rebuttal. As a result, I am willing to increase the score by one point.

---

> > > > ### Author Response · Authors · 2024-08-13
> > > > **Thanks for your review**
> > > >
> > > > Dear Reviewer qa1m,
> > > >
> > > > We are grateful for your thorough review of our submission and the insightful feedback you've provided. Please let us know if you'd like any further information.
> > > >
> > > > Sincerely yours,
> > > >
> > > > Authors.

---

### Author Rebuttal · Authors · 2024-08-07

To all reviewers:

Thank you so much for your careful review and suggestive comments. We have revised our paper according to your comments. The major changes are as follows:

1. We improve the presentation of Sec. 3.1, according to Reviewer qa1m's comments.
2. We add an experiment to evaluate the performance of our model under standard settings, according to Reviewer qa1m's suggestion.
3. We add an experiment to evaluate the robustness of our model *w.r.t.* different base/novel splits, according to Reviewer qa1m's suggestion.
4. We clarify the motivation and working mechanism of our multi-persona collaboration, according to Reviewer qa1m's suggestion.
5. We add an experiment to evaluate the performance when using only one persona, according to Reviewer qa1m's suggestion.
6. We offer more detailed discussions regarding the generation process of descriptions, according to Reviewer qa1m's and xMLQ's comments.
7. We give a detailed discussion regarding the computational complexity, according to Reviewer RK4c's comments.
8. We clarify the usage of possible coexistence descriptions, according to Reviewer xMLQ's comments.
9. We add an ablation study on the effect of integrating possible coexistence descriptions into training, according to Reviewer xMLQ's comments.
10. We discuss the nature of descriptions and CLIP's recognition capability, according to Reviewer xMLQ's comments.

In addition, we include supplementary figures in the PDF of this "global" response for the following aspects:

1. The revised version of Fig. 1c.

Please refer to our response for more details. We have strived to address each of your concerns and welcome further discussions and insights.

Sincerely yours,

Authors.

---

### Decision · Program_Chairs · 2024-09-25

**Decision:**

Accept (poster)

**Comment:**

The reviewers reached a positive consensus early on, and the authors successfully upheld this by effectively addressing their concerns. In their rebuttal, the authors presented several additional experimental results as requested by the reviewers, many of which further strengthen the paper. The AC also strongly recommends including these results in the camera-ready version.